# Geneticin reduces mRNA stability

**Yavuz T. Durmaz**[☯], **Alankrit Shatadal**[☯], **Kyle Friend**[ID]*

Department of Chemistry and Biochemistry, Washington and Lee University, Lexington, Virginia, United States of America

☯ These authors contributed equally to this work.
* friendk@wlu.edu

## Abstract

Messenger RNA (mRNA) translation can lead to higher rates of mRNA decay, suggesting the ribosome plays a role in mRNA destruction. Furthermore, mRNA features, such as codon identities, which are directly probed by the ribosome, correlate with mRNA decay rates. Many amino acids are encoded by synonymous codons, some of which are decoded by more abundant tRNAs leading to more optimal translation and increased mRNA stability. Variable translation rates for synonymous codons can lead to ribosomal collisions as ribosomes transit regions with suboptimal codons, and ribosomal collisions can promote mRNA decay. In addition to different translation rates, the presence of certain codons can also lead to higher or lower rates of amino acid misincorporation which could potentially lead to protein misfolding if a substituted amino acid fails to make critical contacts in a structure. Here, we test whether Geneticin—G418, an aminoglycoside antibiotic known to promote amino acid misincorporation—affects mRNA stability. We observe that G418 decreases firefly luciferase mRNA stability in an *in vitro* translation system and also reduces mRNA stability in mouse embryonic stem cells (mESCs). G418-sensitive mRNAs are enriched for certain optimal codons that contain G or C in the wobble position, arguing that G418 blunts the stabilizing effects of codon optimality.

## Introduction

mRNA stability is a key determinant of protein expression. Thus, cells tightly regulate mRNA stability, often via sequence-specific interactions with mRNA-binding proteins and/or miRNAs. Less specific to individual mRNAs, translation also promotes mRNA decay. Both prokaryotic and eukaryotic translation inhibitors broadly stabilize mRNAs [1–3], and a mutation that disrupts tRNA biogenesis similarly stabilizes mRNAs in yeast [4]. These findings implicate translation as a key determinant of mRNA stability.

More recently, mRNA codon usage has been connected to mRNA stability. Most amino acids are encoded by synonymous codons which often have different usage rates. Within an organism, more prevalent codons are often, but not always, decoded by more abundant tRNAs [5, 6], leading to optimal as well as suboptimal codons whose identities vary between organisms. Over the last decade, it has been observed that optimal codons correlate with increased mRNA stability in both bacterial and eukaryotic systems [7–10]. Since suboptimal codons reduce translational speed in bacteria [11] and are thought to dwell in unoccupied ribosome

**Data Availability Statement:** All high-throughput sequencing files are available from the NCBI Gene Expression Omnibus database (accession number GSE184874).

**Funding:** Our work was funded with financial support from the Thomas F. and Kate Miller

Jeffress Memorial Trust. The Jeffress Memorial Trust did not provide salaries to any of the authors for this study. Please note that the funders had no role in study design, data collection and analysis, decision to publish, or preparation of the manuscript.

**Competing interests:** The authors have declared that no competing interests exist.

acceptor sites (A sites) for longer, these empty A sites might be recognized by some component of the mRNA decay machinery. In fact, unoccupied A sites delay a conformational change in the ribosome, permitting Ccr4-Not complex binding. Due to its role in mRNA decay, binding between the Ccr4-Not complex and ribosomes with unoccupied A sites destabilizes yeast mRNAs with suboptimal codons [12]. In addition, other studies have identified ribosomal stall sites, often with sequential suboptimal codons, as locations where ribosomes collide on mRNAs leading to decay in yeast and mammalian systems [13–15]. In addition to translational optimality, codon nucleotide sequences have been connected to mRNA decay rates. The wobble position, in particular, matters with A/U at the wobble position (AU3) correlating with reduced mRNA stability and GC3 with higher mRNA stability in mammalian cells [16, 17]. These effects may be due in part to tRNA decoding since it is possible to improve translational efficiency either by changing suboptimal codons or cognate tRNAs to improve codon:anticodon base-pairing in yeast [18]. Clearly, both codon optimality and codon sequences play major roles in determining mRNA stability, but we hypothesized that an additional role could be played by amino acid misincorporation.

In addition to effects on translation elongation rates, suboptimal codons are associated with higher bacterial amino acid misincorporation rates [19]. Incorrect codon:anticodon pairing is a common source of bacterial amino acid misincorporation and occurs at higher rates when G:U mismatches can allow for near-cognate tRNAs to bind a codon [20–22]. Near-cognate tRNAs are those tRNAs that can maintain two of three base-pairing interactions during anticodon:codon pairing. Since suboptimal codons typically correlate with lower abundance cognate tRNAs, the ribosome must reject more near-cognate tRNAs while waiting for a cognate tRNA to arrive. This process of tRNA rejection is imperfect, leading to higher amino acid misincorporation rates due to substitution of cognate tRNAs with near-cognate tRNAs [19]. Many of these experiments were potentiated with aminoglycoside translation inhibitors since that class of inhibitor can promote higher rates of amino acid misincorporation [20–22].

Here, we asked whether ribosomal errors affect mRNA stability in mammalian systems. We use G418, an aminoglycoside translation inhibitor that increases amino acid misincorporation rates in mammalian cells [23–25]. By measuring mRNA half-lives in a reporter system and mESCs, we observed that G418 drives mRNA destabilization. *In vitro*, we observe that G418 likely acts independently of ribosome collisions, arguing that its effects are via amino acid misincorporation. *In vivo*, G418 destabilizes mRNAs broadly, in that the majority of mRNAs in mESCs have reduced stability when mESCs are treated with G418. The mRNAs with half-lives that are most reduced by treatment with G418 are enriched for select optimal codons, containing G/C at the wobble position. Together, our results support a potential role for amino acid misincorporation as a regulator of mRNA stability.

## Results

### G418 destabilizes mRNA in rabbit reticulocyte lysate

We hypothesized that amino acid misincorporation events would promote mRNA decay. Since amino acid misincorporation occurs ~1 in 10,000 catalytic cycles, these events could lead to significant background levels of mRNA decay. To investigate our hypothesis, we employed three translation elongation inhibitors, each with unique modes of inhibition. G418 is an aminoglycoside translation inhibitor which promotes amino acid misincorporation and stop-codon readthrough [23–25]. Control translation inhibitors were puromycin and cycloheximide which promote abortive translation and ribosome stalling respectively [26, 27]. First, we titrated the translation inhibitors in rabbit reticulocyte lysate and quantified the levels of firefly luciferase produced from a reporter mRNA (Fig 1A); we identified concentrations of all three

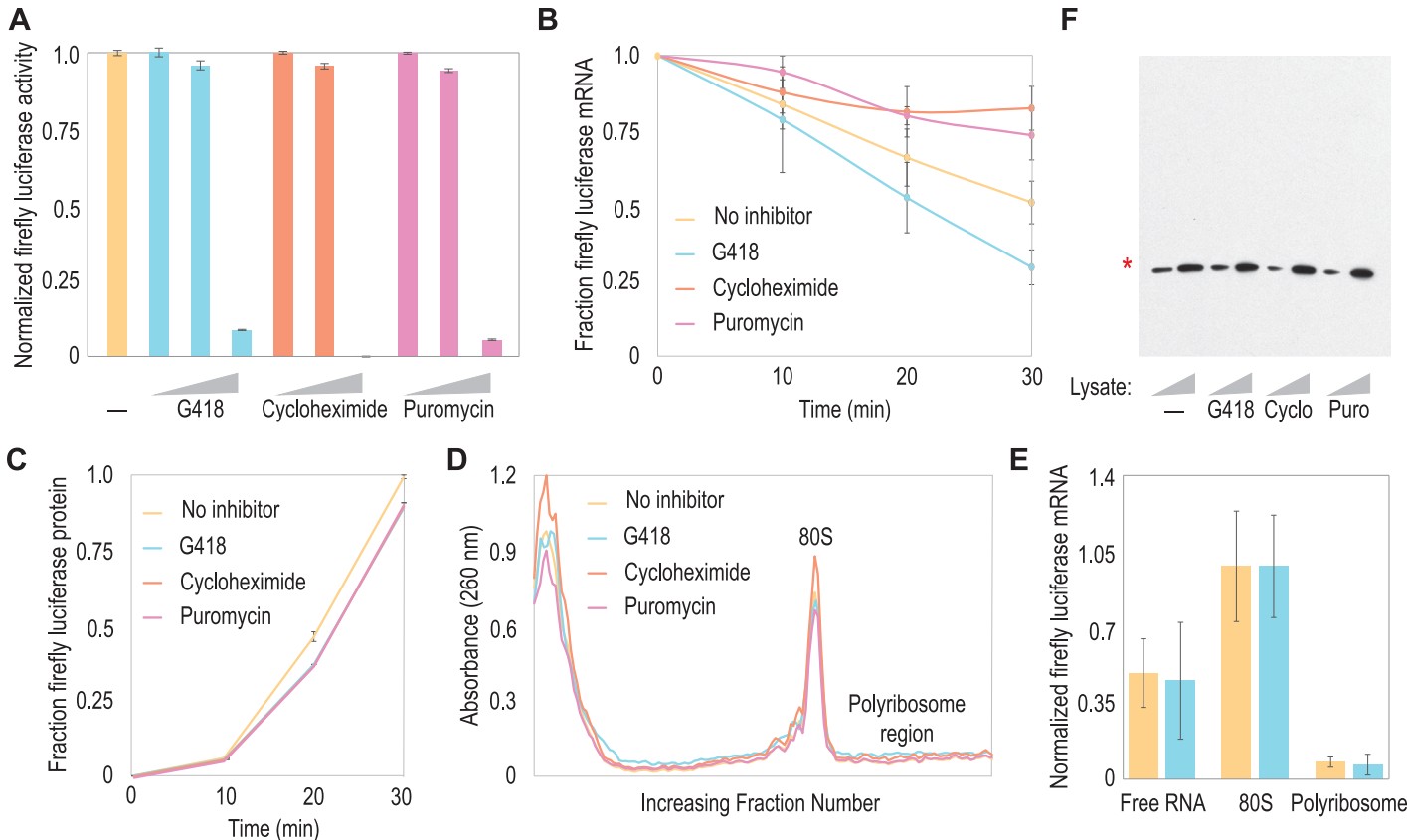

**Fig 1. G418 destabilizes mRNA *in vitro*.** *(A)* Rabbit reticulocyte lysate was used to translate an mRNA encoding firefly luciferase in the presence of G418, cycloheximide, or puromycin at increasing inhibitor concentrations (concentrations were 0.5, 5.0, and 50 ng/μL for G418 and puromycin and 0.25, 2.5 ng/μL, and 25 ng/μL for cycloheximide) to identify concentrations with modest translation inhibition. *(B)* At intermediate antibiotic concentrations (5.0 ng/μL G418 and puromycin, 2.5 ng/μL cycloheximide), G418 destabilizes mRNA relative to control reactions and reactions with other translation elongation inhibitors. After 30 min, *in vitro* translation reactions containing G418 have significantly less mRNA (*, p < 0.01) compared to control reactions, and cycloheximide and puromycin both stabilize mRNA compared to control reactions without inhibitors (*, p < 0.01, p-values from Student's t-test). *(C)* Translation reactions were prepared as in (B), but firefly luciferase protein levels were measured. Very little firefly luciferase production is observed at 10 minutes, but then firefly luciferase accumulates over the remaining time course. Antibiotics consistently reduce luciferase production at all time intervals. *(D)* Again, translation reactions were prepared as in (B) and (C). Reactions were quenched at 15 min and loaded onto a sucrose gradient for ribosome fractionation. Fractions were collected dropwise, and nucleic acid content was quantified at 260 nm. Absorbances for sequential fractions are plotted. Regardless of treatment, a large 80S monoribosome peak was observed with minimal polyribosome peaks. Therefore, the majority of ribosomes in reticulocyte lysate exist as monoribosomes. *(E)* Representative fractions from (D) were probed for firefly luciferase mRNA using RT-qPCR. Consistent with the overall profile in (D), the majority of firefly luciferase mRNA can be found in fractions from the top of the gradient (Free RNA) or the 80S monoribosome peak. Minimal, but detectable firefly luciferase mRNA can be isolated from fractions in the polyribosome region of the gradient. These data suggest that most mRNA in reticulocyte lysate is translated by monoribosomes. *(F)* Proteins from *in vitro* translation reactions were analyzed by western blotting for firefly luciferase. Only full-length protein was visible (*), indicating that G418 treatment did not cause high rates of stop codon readthrough. Due to its mechanism, it is unlikely that treatment with G418 would affect the total protein level as measured here since it promotes amino acid misincorporation. Altogether, these data suggest that G418 destabilizes mRNA.

translation inhibitors where luciferase production was consistently, but modestly depressed. Using these intermediate inhibitor concentrations, we then quantified firefly luciferase mRNA levels during translation reactions. In all cases, ribosomal rRNA was used for normalization. Consistent with our hypothesis, we found that significantly less mRNA was present at the end of translation reactions containing G418 compared to uninhibited reactions (Fig 1B). Consistent with previous research [2, 3], we observed that cycloheximide stabilized mRNA levels relative to uninhibited reactions. The same was true for puromycin. These data suggest that mRNA decay is specific to G418. The contrast between residual mRNA levels and loss of firefly luciferase production was striking in that much more mRNA was degraded compared to lost

protein in reactions treated with G418. Therefore, we performed a time course experiment monitoring firefly luciferase production to assess whether most firefly luciferase was produced at a point before significant levels of mRNA were degraded (Fig 1C). At early time points, very little protein is produced, but between 10 and 30 minutes, there is a rapid accumulation of firefly luciferase protein in control reactions and those treated with antibiotics (Fig 1C). Presumably, the lag in protein production is due to ribosomal loading onto firefly luciferase mRNA and translation elongation through the stop codon. Interestingly, at the antibiotic concentrations in use here, we do not observe a consistent delay in translation elongation by cycloheximide which would be expected given its role in ribosome stalling [27], although 10 minute intervals may not have enough resolution to observe small delays in translation elongation rates. Since G418 might delay translation elongation, we repeated our time course analysis with higher concentrations of all three antibiotics to probe whether G418 could delay elongation (S1 Fig). Under these conditions, both G418 and cycloheximide delay the initial production of firefly luciferase, suggesting both antibiotics delay translation elongation at higher concentrations. An important caveat with this experiment is that higher levels of G418 may compromise firefly luciferase activity, although residual activity still accumulates with delayed kinetics (S1 Fig). G418 is not known to cause ribosome collisions, but since cycloheximide can have this effect via a stalling mechanism [13], we sought to test whether G418 might promote ribosome collisions in our *in vitro* translation system.

During our time course assays, we observed an initial wave of protein synthesis during a more continuous reduction in mRNA levels (compare Fig 1B and 1C). We repeated our *in vitro* translation assays and quenched translation at 15 minutes to overlap with the initial burst in firefly luciferase production while firefly luciferase mRNA levels were decreasing, but not significantly different between reactions (see Fig 1B). We then performed polyribosome sedimentation on these reactions. Polyribosome sedimentation from rabbit reticulocyte lysate has previously been performed, yielding varied results where different groups have observed monoribosomes or polyribosomes engaged in translation [28, 29]. Within our assays, we observe a large monoribosome peak and minimal, if any, polyribosomes lower in the gradient (Fig 1D). It should be noted that we prepared our samples to focus on monoribosome and small polyribosome fractions meaning that our polyribosome sedimentation methodology may exclude very large polyribosome-mRNA complexes. We also isolated RNA and performed RT-qPCR from representative gradient fractions to determine the relative amounts of firefly luciferase mRNA across the gradient. We observed most firefly luciferase mRNA in the monoribosome fraction with very little mRNA in heavier fractions which would correspond to polyribosomes (Fig 1E). These data were consistent with the overall RNA gradient profile. We cannot formally rule out the possibility that G418 promotes ribosome collisions with a rapid loss of a di-ribosome peak, but we do not observe large quantities of di-ribosomes or polyribosomes *in vitro*.

In addition to causing amino acid misincorporation, G418 promotes stop codon readthrough [25]. Since mRNAs with high levels of stop codon readthrough should be degraded by the non-stop decay pathway [30, 31], we tested whether firefly luciferase protein produced in reticulocyte lysate treated with G418 was the proper length. We did not observe detectable levels of extended protein on a western blot (Fig 1F), indicating that minimal stop codon readthrough occurred in translation reactions containing G418. Importantly, G418 drives amino acid misincorporation by the ribosome, and levels of firefly luciferase protein were similar between control reactions and those treated with G418. It is likely that some loss of enzyme activity is due to protein misfolding or loss-of-function due to amino acid substitution. Taken together, our results suggest that G418 can drive higher levels of mRNA decay *in vitro*. We

cannot prove that this effect is independent of ribosome collisions, but our data are more consistent with a role for amino acid misincorporation.

## G418 destabilizes mRNAs in mESCs

Our *in vitro* results confirmed our expectations, but reticulocyte lysate is unusual in that mRNAs are turned over in minutes, rather than hours as has been observed in mammalian cells (discussed in ref. [24]). For this reason, we sought to extend our findings to mESCs. As with our *in vitro* experiments, we first identified translation inhibitor concentrations that would have a modest effect on total protein synthesis. Here, we focused on puromycin as a control translation inhibitor since it stabilized mRNA levels in our *in vitro* experiments (Fig 1B), but it functions similarly to G418 in that it does not stall ribosomes on the mRNA during translation [26]. Using azidohomoalanine to label newly-made proteins and Click chemistry to conjugate a fluorophore onto those newly made proteins [32], we identified intermediate concentrations of G418 and puromycin that modestly inhibited translation in mESCs (Fig 2A).

Next, we sought to globally measure mRNA half-lives in mESCs, and we elected to use SLAM-Seq, a recently published technique that allows pulse-chase analysis with 4-thiouracil which should minimally disrupt protein-RNA interactions and mRNA translation rates [33–35]. Briefly, we pulsed mESCs for one day with 4-thiouracil to accumulate a reservoir of labeled mRNAs and then cultured the mESCs for varied times in the presence of uridine/translation inhibitors for the chase. Since 4-thiouracil mispairs with G after chemical alkylation, U → C conversions are probable sites of 4-thiouracil incorporation that can be detected by sequencing [35]. By quantitating the time-dependent, decreasing fractions of sequencing reads containing U → C conversions, half-lives can be calculated. Under our three growth conditions (control, G418-treated, and puromycin-treated), we were able to determine the half-lives of ~10,300 mRNAs (Fig 2B, S1 Table). Importantly, we observed a destabilizing effect on mRNA half-lives for G418 and a stabilizing effect for puromycin (Fig 2B), consistent with our *in vitro* results (see Fig 1). It should be noted that effects on mRNA stability were modest, but significant. Higher concentrations of the antibiotics might have elicited a more robust difference in mRNA half-lives, but would have a side-effect of significantly disrupting cellular homeostasis due to loss of protein production. Given that mRNA half-life calculations require time points over multiple hours, we elected to use less disruptive inhibitor concentrations, but this may have weakened our observed effects on mRNA half-lives. That being said, G418 does significantly reduce mRNA half-lives, and this is not a general effect of translation inhibitors since puromycin (Fig 2B) and cycloheximide [3] both increase mRNA stability.

Given these initial results, we next correlated mRNA half-lives with codon optimality, both to validate our approach and to ask whether translation inhibition has a mitigating or intensifying effect. Codon Stabilization Coefficients (CSCs) for each codon were calculated using the method in Presnyak *et al*. [7]. Note that positive CSCs indicate a stabilizing effect, and negative CSCs indicate a destabilizing effect. When organized by CSC score, neither G418 nor puromycin significantly changes CSC values for all codons (Fig 2C), but rather individual correlations are often slightly shifted by the antibiotics. Separately, there is a general trend (with some clear exceptions) between increasing codon optimality and increasing mRNA half-lives consistent with prior publications (see S2 Table and refs. [4–7]), although it should be emphasized that the tRNA adaptation index (tAI) which we use as a metric for codon optimality [5] can vary depending on cellular growth conditions [36].

We next sought to identify a group of mRNAs whose half-lives most responded to G418. We compared mRNA half-lives from G418-treated cells to either control or puromycin-treated cells, focusing on mRNAs whose half-lives progressively decreased when comparing

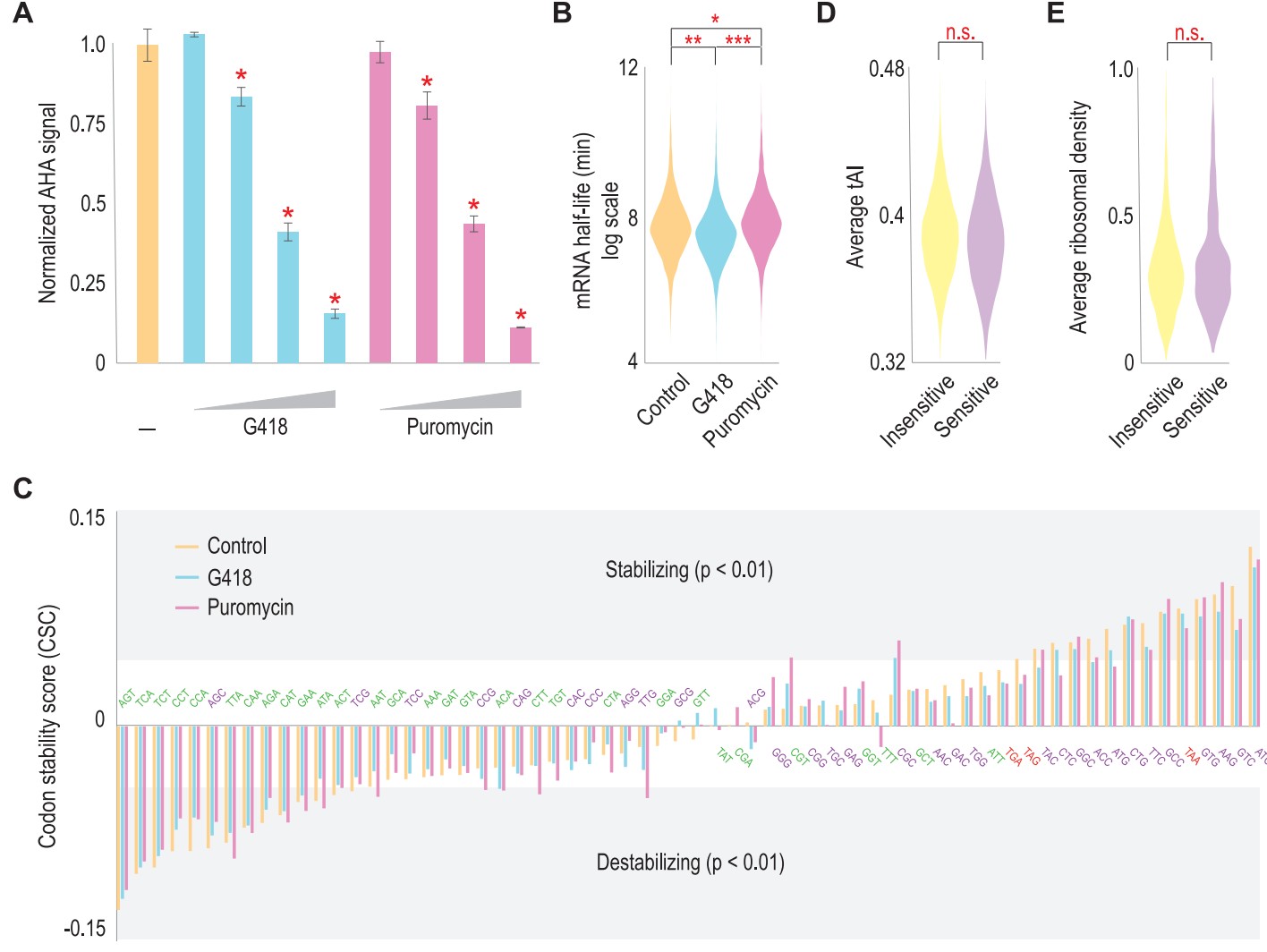

**Fig 2. Amino acid misincorporation drives mRNA instability in mESCs.** *(A)* mESCs were treated with increasing concentrations of G418 and puromycin in the presence of azidohomoalanine (AHA). AHA incorporation was monitored by fluorescence after conjugating AlexaFluor 488 to AHA using Click chemistry. At the indicated concentrations (*), G418 and puromycin both significantly depressed new protein synthesis ($p < 0.01$, Student's t-test). *(B)* SLAM-seq analysis was used to calculate mRNA half-lives in mESCs, comparing control cells to those grown in the presence of G418 (higher amino acid misincorporation rates) and puromycin (abortive translation elongation). Shown are violin plots for ~10,600 mRNA half-lives in the three conditions. P-values (*, p = 1.1 e-5, **, p = 3.3 e-16, and ***, p = 8.9 e-36) were calculated using the Mann-Whitney U-test. *(C)* Shown are codon stability scores for the fraction of codons in an mRNA correlated with mRNA half-lives. Positive correlations mean that an amino acid codon is more likely to be present in a stable mRNA (Stabilizing) and vice-versa (Destabilizing). Codons are arranged by increasing CSCs for mESCs grown under control conditions, and codon sequences are given in the graph with coloring according to wobble position nucleotide (green are AU3 codons, and purple are GC3 codons). In all cases, mESCs treated under various conditions had similar correlation coefficients, and there is a general trend with stabilizing GC3 codons. *(D)* mRNAs were divided into G418-sensitive (Sensitive) mRNAs and all other mRNAs (Insensitive) by calculating the ratio between mRNA half-lives in G418-treated versus puromycin-treated mESCs (see Materials and Methods). Then average codon optimality (CSC score) was calculated for each group. There is no significant difference in CSC scores between groups. *(E)* Similarly, ribosome density (from ref. [28]) was compared for G418-sensitive mRNAs to all other mRNAs. We again observed no difference in average ribosome density.

puromycin-treated to control and then G418-treated mESCs (see Materials and Methods). In analyzing G418-sensitive mRNAs, we do not observe a statistically significant difference in codon optimality between G418-sensitive mRNAs and the remaining mRNAs (Fig 2D). Together with Fig 2C, these data confirm the role of individual codons in regulating mRNA stability in mESCs, but also suggest that translation inhibitors regulate mRNA stability via an independent mechanism. The use of translation inhibitors might also suggest that

G418-sensitive mRNAs were simply more heavily translated, but this was not the case. Global mRNA translation levels have already been measured in mESCs [37]. Using those data, we confirmed that G418-sensitive mRNAs do not have higher ribosome density compared to insensitive mRNAs (Fig 2E). We also performed Gene Ontology analysis and analyzed G418-sensitive mRNA lengths, neither of which yielded strong differences between G418-sensitive mRNAs and the remaining mRNAs (most significant Gene Ontology category: pre-mRNA splicing, p = 0.0018; mRNA lengths: sensitive—2067 nt, insensitive—2064 nt, p = 0.22). In summary, we do identify a population of G418-sensitive mRNAs, but they are not characterized by differences in codon optimality, ribosome density, length, or encoded protein function.

Since codon nucleotide sequences have been correlated with mRNA half-lives, we also sought to correlate codon sequences within mRNA half-lives. Wobble position nucleotides can have stabilizing (GC3) or destabilizing (AU3) effects on mRNA stability [16], so we analyzed all three codon positions for potential stabilizing or destabilizing effects on mRNA half-lives (Fig 3A). Our results are consistent with Hia *et al*., although we do see greater effects from specific nucleotides such as A in the wobble position, which correlates with greater mRNA instability compared to U. As with codon optimality, G418 and puromycin shift the correlations between wobble position nucleotides and mRNA stabilities, but the effect is modest.

So, how do G418 and puromycin act? Both significantly change mRNA half-lives, with G418 destabilizing mRNAs and puromycin stabilizing mRNAs (Fig 2B). Since G418 may act at the level of amino acid misincorporation, we hypothesized that certain codons, and thus certain amino acids, would be more sensitive to G418. Rather than focus on all mRNA half-lives, we separated out the G418-sensitive mRNAs for further analysis as above. In doing so, we observed an intersection between codon optimality and nucleotide preferences. All codons enriched in G418-sensitive mRNAs contained G or C in the wobble position, and some were optimal (see Fig 3B and S3 Table). In addition to a G or C in the wobble position, G418-sensitive mRNA codons also contained another G or C in the first or second position. Among codons that were underrepresented in the G418-sensitive mRNAs, the plurality encoded a hydrophobic amino acid with the exceptions being those encoding Asn and Asp. The underrepresented codons for Asp and Asn are known sites of amino acid misincorporation [38]. These codons are likely already destabilizing leading G418 to have a weaker effect on these mRNAs. One caveat with these analyses is the lack of a reporter gene with GC3 or AU3 codons to directly assess the role of G418 and puromycin in regulating mRNA stability, but these would be interesting future analyses. As mentioned above, we do observe that most G418-sensitive mRNAs are enriched for codons with G or C in the wobble position. The structure of a yeast ribosome bound to G418 is solved, and it was observed that G418 promoted near-cognate tRNA accumulation within the ribosomal A site [39]. Prokhorova *et al*. did not systematically check all A site tRNA:codon pairs, but it is tempting to speculate that G418 may preferentially allow near-cognate tRNA usage with codons containing greater GC content. Ultimately, if near-cognate tRNAs are used in translation, it would be expected to disrupt protein folding. Consistent with a model where protein misfolding may connect to G418's mode of action, cells treated with G418 are known to contain higher concentrations of protein aggregates [40] and have induced ER stress pathways [41], suggesting that G418 may drive protein misfolding. It is important to note that our *in vitro* experiments do show that G418 can delay translation elongation, potentially leading to ribosome collisions *in vivo*. Here, we cannot formally rule out this possibility, but our *in vitro* assays would favor a model where G418 acts via an independent mechanism, likely at the level of amino acid misincorporation. Taken together, G418 preferentially dampens the protective role of codons containing G/C in the wobble position.

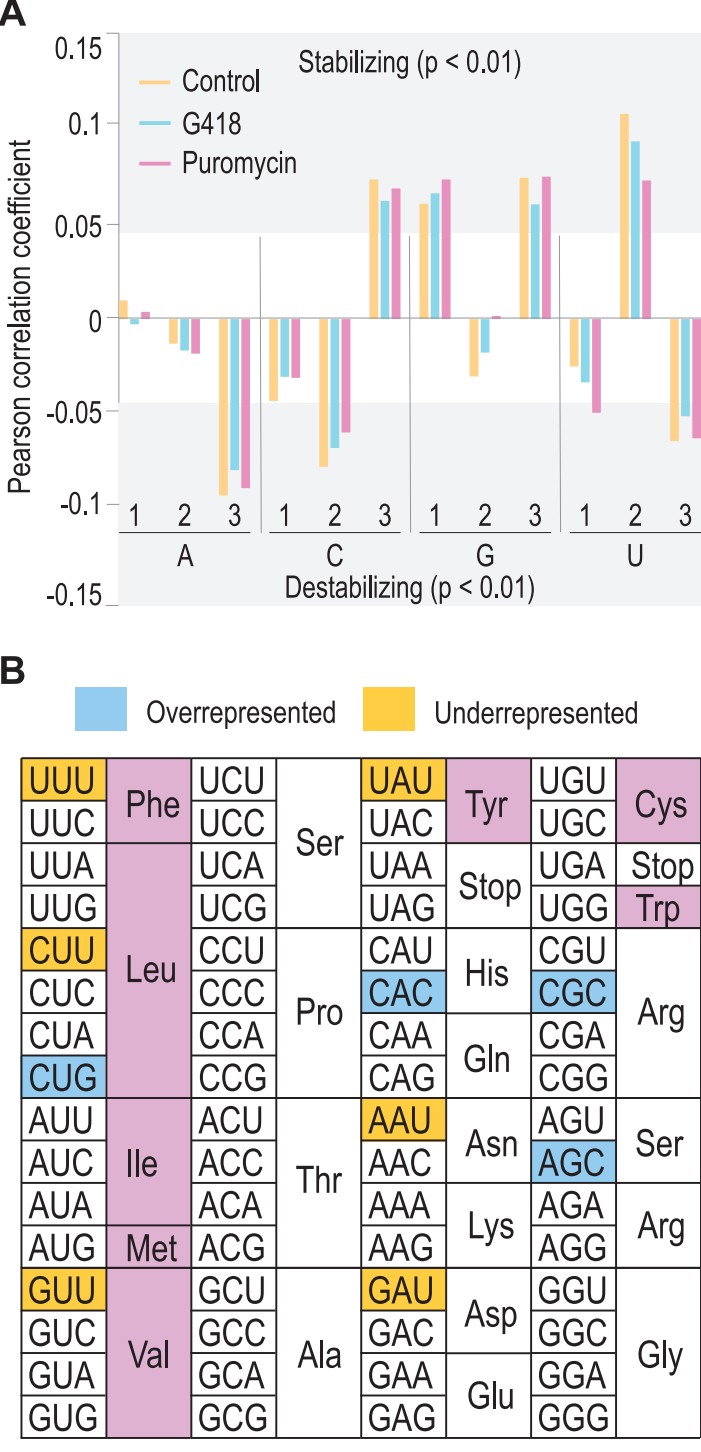

**Fig 3. G418 preferentially destabilizes mRNAs with G/C nucleotides in the wobble position.** *(A)* We calculated the fraction of nucleotides at each codon position and correlated those fractions with mRNA half-lives. Consistent with published results [16], we observed destabilizing effects if the wobble position was occupied with either an A or U. Treatment with G418 or puromycin yielded results that were consistent with control mESCs. *(B)* We analyzed individual codons to see which were overrepresented or underrepresented in the pool of G418-sensitive mRNAs. Many of the codons with a U in the wobble position and encoding hydrophobic amino acids were underrepresented in the G418-sensitive mRNAs. Additionally, AU3 Asp and AU3 Asn codons were found. These did not consistently align with suboptimal codons. Since the Asp and Asn codons are known to have higher rates of amino acid misincorporation [38], they may act as a sensitized background to observe effects of amino acid misincorporation.

Overrepresented codons contained GC3 in the wobble position. Codons were only labeled as overrepresented or underrepresented if p < 0.01 (from a Mann-Whitney U-test).

## Discussion

To study the connection between translation dynamics and mRNA stability, we used different translation inhibitors with separate modes of action to alter global mRNA stability. Depressing translation elongation with cycloheximide or puromycin leads to enhanced mRNA stability, but targeting the ribosome with an aminoglycoside that drives higher rates of amino acid misincorporation promotes mRNA decay. We observe these effects *in vitro* as well as in mESCs. *In vitro*, G418 destabilizes mRNAs that are largely bound by monoribosomes, but can depress translation elongation rates. Our observations are more consistent with a model where G418 operates via amino acid misincorporation, but we cannot exclude the possibility that G418 promotes ribosome collisions. By examining G418-sensitive mRNAs, we observe an enrichment of codons that terminate in a G or C in the wobble position with a concomitant reduction in codons terminating with A or U in the wobble position. Since GC3 codons are often associated with enhanced mRNA stability and AU3 codons with reduced mRNA stability, G418 dampens codon effects at the wobble position.

Based on our findings, a key question is why G418 destabilizes mRNAs that are enriched with select GC3 codons. In mESCs, these may be less readily translated. It has been observed that proliferating cells differentially express tRNAs compared to nonproliferating cells [36]. In particular, proliferating cells are enriched for tRNAs that decode AU3 codons whereas non-proliferating cells express higher concentrations of tRNAs that decode GC3 codons [36]. Since our mESCs were cultured to maintain high rates of proliferation, we would expect GC3 codons to be less readily translated. That would increase the probability of near-cognate tRNAs outcompeting cognate tRNAs and could serve as a more sensitive background in which G418 could act. Given that aminoglycoside antibiotics deform the decoding center at the wobble position [39], that might explain the nucleotide bias we observe in G418-sensitive mRNAs.

How might G418 promote mRNA instability? *In vitro*, we observe that G418 likely acts on mRNAs that are bound to single ribosomes. These observations would suggest a connection between protein misfolding and mRNA decay since G418 is known to drive higher error rates in the ribosome. For some time, it has been known that protein misfolding can be coupled to mRNA instability under specific circumstances. When signal sequences are altered, secreted or membrane protein-encoding mRNAs are rapidly and efficiently degraded by the regulation of aberrant protein production pathway [42, 43]. We do not observe that G418-sensitive mRNAs are enriched for secreted or membrane proteins, suggesting an additional cytosolic mechanism connecting protein misfolding to mRNA decay. Ubr1 is a ubiquitin ligase that co-translationally recognizes misfolded proteins and leads to their ubiquitination [44, 45]. It is not known to directly regulate mRNA decay, but it was identified in a complex with one of the major deadenylases in the cell, Ccr4, in a yeast high-throughput screening assay [46]. It is tempting to speculate that G418-sensitive mRNAs are degraded by this or a similar pathway. Importantly, we do observe that G418 can delay translation elongation, and G418 destabilizes mRNAs *in vivo*. In this setting, it is certainly possible that G418 promotes ribosome collisions which are known to destabilize mRNAs [13–15]. Perhaps ribosome collisions and amino acid misincorporation act synergistically to destabilize mRNAs. This would be an interesting future research question. In summary, we show that G418 treatment leads to mRNA instability, with an implied connection between codon identity and mRNA decay.

## Materials and methods

### In vitro translation and ribosome sedimentation

18 μL of nuclease-treated rabbit reticulocyte lysate (Promega) was incubated with 0.5 μg of the supplied firefly luciferase mRNA and 1 μL of 1 mM amino acids in a final volume of 20 μL. Where indicated, antibiotics were added at 5.0 ng/μL puromycin, 2.5 ng/μL cycloheximide, and 5.0 ng/μL G418. For reactions with higher concentrations of antibiotics, 50 ng/μL puromycin, 25 ng/μL cycloheximide, and 50 ng/μL G418 were used respectively. Reactions with lower concentrations of antibiotics had 0.5 ng/μL puromycin, 0.25 ng/μL cycloheximide, and 0.5 ng/μL G418. For antibiotic titration experiments, reactions were incubated for 30 min at 30 ˚C, and for time course experiments, reactions were incubated for the indicated times at 30 ˚C. In all cases, a zero time point control sample was also prepared and placed on ice. Half the reaction volume was used to determine firefly luciferase protein expression, and the remaining volume was used for RT-qPCR as outlined in the next paragraph. For firefly luciferase protein expression, 40 μL of pre-warmed Luciferase Assay Substrate (Promega) was added. Luminescence was monitored on a Tecan M1000 Pro microplate reader. Background luminescence was calculated by averaging the zero time point samples and subtracted from each non-zero time point. Statistical significance was determined using a Student's t-test.

For ribosome sedimentation, *in vitro* translation reactions were performed as indicated above. Where indicated, antibiotics were added at 5.0 ng/μL puromycin, 2.5 ng/μL cycloheximide, and 5.0 ng/μL G418. After 15 minutes of incubation, translation reactions were quenched with ribosome homogenization buffer (10 mM Tris, HCl, pH 7.5, 1.5 mM $MgCl_2$, 10 mM KCl, 2 mM DTT, and 100 ng/μL cycloheximide). We quenched reactions at this time point since it corresponded to a period between first production of firefly luciferase and a large burst of firefly luciferase production at 20 minutes while mRNA decay was ongoing. Reactions were then overlaid onto a 10%—50% sucrose step gradient (10 mM Tris, HCl, pH 7.5, 1.5 mM $MgCl_2$, 10 mM KCl, 2 mM DTT with 10%, 20%, 30%, 40%, or 50% sucrose w/v). Gradients were then centrifuged at 39,000 rpm for 3 hrs in a SW41 rotor. Fractions were collected dropwise into the wells of a 96-well plate after puncturing the bottom of the polyallomer centrifuge tube with an 18G needle. Absorbance was then quantified on a NanoDrop spectrophotometer.

### RT-qPCR

For RT-qPCR, first total RNA was prepared from the *in vitro* translation reactions or ribosome sedimentation fractions. In both cases, 200 μL of G25 buffer was added (300 mM NaOAc, 1% SDS, 10 mM Tris, and 1 mM EDTA with pH adjusted to 7.5). For ribosome sedimentation experiments, it was necessary to normalize to a spike-in mRNA control. 20 fmol of *in vitro* transcribed CFP-encoding RNA was added to provide a normalization control. Samples were mixed and extracted with 300 μL PCA (phenol:chloroform:isoamyl alcohol, 25:24:1). To the supernatant, 1 μL of 5 mg/mL glycogen and 2.5 volumes of ethanol were added. Samples were incubated at -80 ˚C and pelleted at 14,000 rpm for 15 min. Pellets were washed with 100 μL of ice-cold 70% ethanol and dried. Dried RNA pellets were then resuspended in 10 μL TE buffer (10 mM Tris, 1 mM EDTA with pH adjusted to 7.5). 500 ng of RNA was used for reverse transcription with random nonamers (Sigma) and MMLV reverse transcriptase (Invitrogen) according to the manufacturer's protocol. For the RT reaction, RNA, water and primers were pre-incubated at 25 ˚C for 10 min; the remaining reverse transcriptase mixture was added; and reactions were incubated at 42 ˚C for 1 hr. After reaction, 0.5 μL of RNase H (Invitrogen) was added, and reactions were incubated for 15 min at 37 ˚C. qPCR reactions were prepared using iTAQ Universal SYBR Green Supermix (Bio-Rad) and gene-specific primers (rabbit 18S rRNA

primers (3′ end): CCAAATGTCTGAACCTGCGG and GTGAAGCAGAATTCACCAAGC, firefly luciferase primers (in CDS): TCTTGCGTCGAGTTTTCCGG and GCACGGAAAGACGATGACGG, CFP primers (in CDS): AGATGCCACGTACGGGAAAC and AATCGTGCTGTTTCATGTGG). qPCR reactions were monitored on a Bio-Rad CFX Connect Real-Time System (Bio-Rad) with a 56 ˚C annealing temperature. Quantitation was performed by the $\Delta\Delta C_q$ method. As above, statistical significance was determined using a Student's t-test.

## Western blotting

*In vitro* translation reactions were prepared as indicated above. Where indicated, antibiotics were added at 5.0 ng/μL puromycin, 2.5 ng/μL cycloheximide, and 5.0 ng/μL G418. Reactions were then separated on an 8% SDS-PAGE gel and western blotted for firefly luciferase (Mouse monoclonal antibody, CS 17, Invitrogen).

## Azidohomoalanine labeling

mESCs were cultured in methionine-free, ESGRO 2i medium (Millipore) with antibiotics (1, 0.5, 0.1, and 0.05 μg/mL puromycin or 1, 0.5, 0.1, and 0.05 μg/mL G418) for 4 hrs. Azidohomoalanine was added at 25 μM to the medium during the incubation period [35]. mESCs were harvested as above, and cell pellets were resuspended in lysis buffer (50 mM Tris, 0.1% SDS with pH adjusted to 8.0). Proteins were labeled with Alexa 488-alkyne and the Click-iT Protein Reaction Buffer Kit according to the manufacturer's protocol (Thermo Fisher). After reaction, proteins were precipitated with two volumes of ice-cold acetone, and pellets were resuspended in 100 μL PBS containing 8 M urea. Fluorescence was measured in a microplate reader (M1000 Pro, Tecan).

## mESC culture and SLAM-Seq

Mouse embryonic stem cells (E14Tg2a, ATCC) were cultured in ESGRO-2i medium (Millipore) to maintain pluripotency with daily medium exchanges under standard growth conditions [47]. When appropriate, mESCs were passaged using ESGRO Complete Accutase (Millipore) according to manufacturer's instructions. 100 μM 4-thiouracil (Sigma) was incubated with the cells over a 24 hour period as described [35] except medium was exchanged every 8 hours. Medium containing antibiotics (0.1 μg/mL puromycin or 0.1 μg/mL G418) along with uridine (at 10 mM, Sigma) was added during the chase period, and cells were harvested at various time points (0, 1, 2, 4, 8, and 24 hrs after medium change) with two replicates per sample. For harvesting, mESCs were first washed one time in PBS and then treated with ESGRO Complete Accutase (Millipore) before centrifugation at 500 g for 2 min. Cell pellets were washed 2 times in PBS, and the pellets were frozen in liquid nitrogen for storage.

For SLAM-Seq analysis, Trizol reagent was used to prepare total RNA from mESCs labeled with 4-thiouracil according to the manufacturer's instructions (Invitrogen). RNAs were then alkylated as previously described using iodoacetamide [35]. Paired end RNA sequencing was then performed by Genewiz using an Illumina HiSeq 2000. Sequencing reads were then aligned to the mouse transcriptome using the Bowtie2 algorithm [48] and the mm10 reference genome. Once aligned, custom scripts were used to analyze alignment files for U → C conversion, and the fraction of reads containing converted U were calculated for each mRNA (see custom scripts). The fractions of labeled transcripts were then fit to an exponential decay curve to calculate mRNA half-lives.

## Half-life analysis

Again using custom scripts, we first calculated codon optimality for every mRNA in our dataset according to ref. [7]. Using the Scipy package, we then calculated a Pearson correlation coefficient between codon optimality and mRNA half-lives. For nucleotide position analysis, we calculated the fraction of codons containing a specific nucleotide at each of the three codon positions, and as above, we calculated Pearson correlation coefficients for these fractions and mRNA half-lives.

To identify the G418-sensitive mRNAs, we calculated the ratio of G418-treated mESC mRNA half-life to puromycin-treated half-life for every mRNA. mRNAs that had half-lives two-fold lower in G418-treated mESCs relative to puromycin-treated cells were further evaluated. Those mRNAs whose half-lives were greatest in puromycin-treated cells, an intermediate value in control cells, and lowest in G418-treated cells were labeled as G418-sensitive. In analyzing these mRNAs for codon optimality, codon composition, and codon nucleotide sequences, we used the Mann-Whitney U-test to determine p values comparing G418-sensitive mRNAs to the remaining mRNAs.

## Supporting information

**S1 Fig. G418 can delay translation elongation.** *(A)* Rabbit reticulocyte lysate was used to translate an mRNA encoding firefly luciferase in the presence of high concentrations of translation inhibitors (50 ng/μL G418, 25 ng/μL cycloheximide, or 50 ng/μL puromycin). Translation reactions were incubated for the indicated time points, and firefly luciferase protein levels were measured by luminescence. Very little firefly luciferase production is observed at 10 minutes, but then firefly luciferase accumulates over the remaining time course. Antibiotics consistently reduce luciferase production at all time intervals, but G418 and cycloheximide both delay the onset of firefly luciferase protein production. This is best observed in *(B)* where control reaction data are removed from the plot. At 10 min, almost no firefly luciferase protein is observed in reactions with G418 or cycloheximide, but reactions with puromycin do exhibit firefly luciferase protein. By 30 minutes, all three inhibitors yield similar levels of firefly luciferase protein. These data suggest G418 and cycloheximide detectably delay translation elongation at higher inhibitor concentrations.
(EPS)

**S1 Table. SLAM-Seq mRNA half-lives.** mRNA half-lives from the SLAM-Seq protocol are reported. Transcript identifiers are given along with mRNA half-lives calculated according to ref. [35]. mESCs were cultured either under control conditions (ESGRO 2i medium, Millipore) or in ESGRO 2i medium containing 0.1 μg/mL G418 or puromycin with half-lives calculated in minutes. All results were determined using two biological replicates, corresponding to 6 total samples. The average number of sequencing reads used to calculate each mRNA half-life are given in the final column.
(XLSX)

**S2 Table. Codon stability scores for individual codons.** Individual codons are listed with calculated CSC scores for mRNA half-lives determined using the SLAM-Seq protocol. mESCs were cultured under control conditions or were treated with 0.1 μg/mL G418 or puromycin as indicated. Note that negative values indicate codons that are destabilizing whereas positive values indicate codons that are stabilizing. Codon adaptation indices are given to reflect codon optimality where low values indicate suboptimal codons, and higher values indicate more optimal codons. Lastly, the wobble position nucleotide is separated out to show that most codons

with negative CSC values end in A or U, whereas stabilizing codons more often end in G or C.
(XLSX)

**S3 Table. G418-sensitive mRNAs are enriched for specific codons.** The raw data are provided corresponding to Fig 3B. The average percentage of individual codons are given in the G418-sensitive and insensitive mRNAs. A Mann-Whitney U-test was used to compare the distribution of codon percentages in the G418-sensitive and insensitive mRNAs, with p-values indicated.
(XLSX)

## Author Contributions

**Conceptualization:** Yavuz T. Durmaz, Kyle Friend.

**Data curation:** Alankrit Shatadal, Kyle Friend.

**Formal analysis:** Yavuz T. Durmaz, Alankrit Shatadal, Kyle Friend.

**Funding acquisition:** Kyle Friend.

**Investigation:** Yavuz T. Durmaz, Alankrit Shatadal.

**Methodology:** Yavuz T. Durmaz, Alankrit Shatadal.

**Project administration:** Kyle Friend.

**Supervision:** Kyle Friend.

**Writing – original draft:** Alankrit Shatadal, Kyle Friend.

**Writing – review & editing:** Yavuz T. Durmaz, Alankrit Shatadal, Kyle Friend.

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
