## [Decision Letter · Decision Letter 0]

17 Feb 2022

PONE-D-22-00848Amino acid misincorporation reduces mRNA stabilityPLOS ONE

Dear Dr. Friend,

Thank you for submitting your manuscript to PLOS ONE. After careful consideration, we feel that it has merit but does not fully meet PLOS ONE’s publication criteria as it currently stands. Therefore, we invite you to submit a revised version of the manuscript that addresses the points raised during the review process.

Below you will find evaluation of three reviewers who all agree that the manuscript presents experimental data that addresses an important question and will be a valuable contribution. However, the reviewers also express significant concerns about the experimental approach, data analysis, data presentation, data interpretation and overall conclusions. We invite you to revise your manuscript in light of these comments. Here are some key points that should be addressed in the revision:

As reviewers also emphasize, there is no direct evidence provided in the manuscript for link between amino acid misincorporation and mRNA stability. The observations described can be explained via alternative models. Thus, the current title and undue focus on this one mechanism is unwarranted. Either more direct evidence will be needed to support the current model, or the title and text should be extensively revised to present a more balanced interpretation of data reported in the manuscript. In any scenario, various possible modes by which translation can impact mRNA stability should be discussed.

New experiments have been suggested to obtain data for more direct support of the conclusions regarding G418 effects on mRNA stability in a codon dependent manner (reviewer 1, comments 2 and 3; reviewer 2, major comment 2). One possible way to address this could be to perform new experiments focused on select G418 sensitive and insensitive RNAs. Another possibility is to acknowledge the lack of direct evidence for such a mechanism and extensively revise the conclusions while clearly stating limitations of the data and possible alternate explanations for the observations.

Both reviewer 1 and 2 point out the need to address the discrepancy between small effect on luciferase activity but significant impact on reporter RNA stability by intermediate levels of G418. Reviewer 2 suggests a possible source of this discrepancy that can be experimentally tested. Another way to address this issue could be to note this discrepancy with a discussion of possible caveats and limitations of the experimental approach.

Both reviewers 2 and 3 express concerns about data analysis and data presentation. All comments concerning these issues should be addressed with more thorough analysis of the available data and its appropriate presentation so that data on which conclusions are based is readily accessible to readers.

In addition, all other comments should be carefully considered to revise the manuscript accordingly. A point-by-point response to reviewer’s concerns should be included in the revised submission.

We look forward to receiving your revised manuscript.

Kind regards,

Guramrit Singh

Academic Editor

PLOS ONE

Journal Requirements:

"We thank the Jeffress Memorial Trust for funding this research (to K.F.). Y.D., A.S., and K.F. designed and performed experiments. A.S. and K.F. wrote this manuscript with support from Y.D."

We note that you have provided funding information. However, funding information should not appear in the Acknowledgments section or other areas of your manuscript. We will only publish funding information present in the Funding Statement section of the online submission form. 

"K. Friend

Jeffress Memorial Trust

https://hria.org/tmf/jeffress/

NO"

Reviewers' comments:

Reviewer's Responses to Questions

**Comments to the Author**

1. Is the manuscript technically sound, and do the data support the conclusions?

Reviewer #1: Partly

Reviewer #2: Partly

Reviewer #3: Partly

2. Has the statistical analysis been performed appropriately and rigorously? 

Reviewer #1: Yes

Reviewer #2: Yes

Reviewer #3: Yes

3. Have the authors made all data underlying the findings in their manuscript fully available?

Reviewer #1: Yes

Reviewer #2: Yes

Reviewer #3: Yes

4. Is the manuscript presented in an intelligible fashion and written in standard English?

Reviewer #1: Yes

Reviewer #2: Yes

Reviewer #3: Yes

5. Review Comments to the Author

Reviewer #1: The article titled “Amino acid misincorporation reduces mRNA stability” by Durmaz et al., addresses an important aspect of RNA Biology. Earlier reports suggest the role of codon optimality and codon sequences in determining mRNA stability. Here the authors tried to suggest a hypothesis where the mRNA stability is affected by misincorporation of amino acid during the process of translation. In this article the authors mimicked the process of amino acid misincorporation using G418 (Geneticin). They further hypothesize that misincorporation of amino acid by ribosome leads to misfolded protein response resulting in protein degradation, where authors wanted to connect this process with mRNA stability.

The article addresses an interesting question and can be considered for publication. However the following issues need to be addressed.

Major comments:

1. G418 treatment reduced the stability of firefly luciferase mRNA in vitro (Figure 1 B). But it did not affect the protein level of firefly luciferase in Figure 1C. It is unclear why this would be the case and authors did not comment about the same.

2. To strengthen the hypothesis proposed by author connecting G418 treatment and mRNA stability, it would be important to test and validate the stability of couple of G418 sensitive mRNA targets with specific codon features.

3. To confirm that amino acid mis-incorporation leads to ribosome slow-down and affect mRNA stability. An important experiment could be to express firefly luciferase mRNA containing optimal codons and non-optimal codons in vivo, check for its translation and stability.

Reviewer #2: In this manuscript, Durmaz YT, Shatadal A and Friend K, studied the impact of different translation inhibitors on modulating mRNA stability in vitro and in mouse embryonic stem cells.

Using an in vitro translation system programmed with a luciferase reporter mRNA in the presence of G418 (an aminoglycoside that alters translation elongation and termination), cycloheximide (which interferes with the translocation step and blocks elongating ribosomes on the mRNA) or puromycin (an aminonucleoside that induces premature chain termination, releasing elongating ribosomes from the mRNA), the authors measured transcript stability in a 30 minutes time course. Their results indicate a significant stabilization of the reporter mRNA in the presence of cycloheximide and puromycin (compared to the untreated control), while G418 leads to accelerated mRNA degradation (compared to the untreated control).

Slam-seq experiments performed in mouse embryonic stem cells validate their in vitro results, showing a global stabilization of cellular transcripts when cells are incubated in puromycin, while incubation with G418 is associated with transcript destabilization. Calculation of codon stabilization coefficients (CSCs) using the Slam-seq datasets from control cells also recapitulates previous findings obtained in other cell-types and in particular the observed GC content bias for stabilizing and destabilizing codons described in Hia et al 2019. Interestingly, addition of puromycin or G418, although having an impact on global mRNA stability, does not appear to significantly affect the CSCs values, even though a small effect is apparent on most codons.

Finally, analysis of codon usage on G418-sensitive mRNAs revealed an enrichment of codons with G or C in the wobble position, which are typically stabilizing codons. Furthermore, codons under-represented in the G418-sensitive mRNAs correspond to hydrophobic amino-acids.

Based on their results and in ribosome crystal structures obtained in the presence of G418, the authors conclude that G418 could preferentially affect near-cognate tRNA usage in GC rich codons leading to increase amino-acid mis-incorporation and inducing mRNA degradation.

Overall, the manuscript is very well written. The introduction is well documented and relevant to the study as it allows readers to place the study in the current context of the field while highlighting open questions that have not been addressed yet. The authors have generate an interesting dataset using the state-of-the-art Slam-seq protocol. However, the analyses performed on this dataset are sometimes cryptic and superficial and could be largely improved. Moreover, there is room for improving the overall clarity of the figures to facilitate the interpretation of the results by readers.

Below you will find some major and minor points that, in my opinion, should be addressed by the authors.

Major points:

- I am surprised that the low concentrations of translation inhibitors used in the in vitro time course (Figure 1B), have such a profound impact on the reporter mRNA stability while leading to a decrease of less than 10% of the luciferase activity at 30 minutes as shown in Figure 1A. This could be due to the accumulation of most of the luciferase protein during the first minutes of the translation reaction followed by a plateau in its abundance at later points. It would therefore be important for the authors to include a new time-course figure where firefly luciferase is measured for each sample at each time-point so readers can compare the dynamics of mRNA levels and protein accumulation.

- Authors decided to perform the Slam-seq protocol in the presence of small amounts of G418 or Puromycin which have modest effects on total translation rates (20% inhibition based on the results presented in Figure 2A). It is therefore not surprising that the effects obtained are very mild. Although translation is an important determinant of mRNA stability, it is probable that only a small fraction of all translating ribosomes are involved in inducing mRNA degradation. It would be interesting to include an additional experiment using a higher concentration of G418 in the analysis to obtain a clearer and robust picture of the impact of G418 on codon-dependent mRNA stability. I also think it would be important to perform a sucrose gradient experiment at the different doses of translation inhibitors to look at their effect on the polysome profile.

In any case, contrary to what it is stated in the results section, I do not think (at least for puromycin) that the effect of the translational inhibitors on modulating mRNA stability are independent from codon optimality. There is robust evidence from the work of Olivia Rissland and Ariel Bazzini laboratories showing that translation inhibition leads to a loss of the effect of codon-optimality on mRNA stability.

- Figure 2B: Instead of displaying a bar plot with the half-lives of cellular transcripts, it would be preferable to use a density plot so readers can have a clearer view of the distribution of half-lives in each condition.

- G418-sensitive transcripts were obtained by calculating the ratio of mRNA half-lives in the G418 treated samples against Puromycin-treated samples (as described in the Material and Methods section). I supposed this was done to maximize the difference in half-lives since the effect of each drug are mild compared to the control condition. However, this strategy might introduce a bias as puromycin also has a small effect on CSC values which are different from that of G418. In my opinion it would be preferable to use the control condition to identify G418-sensitive transcripts. I also think that authors should show a plot with the distribution of the obtained ratios and the two-fold cutoff chosen to define G418-sensitive transcripts.

- Are G418-sensitive transcripts enriched in a specific functional category? Are transcripts with the longer open reading frames more sensitive to G418 since they will potentially accumulate more mis-incorporated amino-acids in their nascent chains? Is the Ribosome-associated quality control machinery recruited to polysomes upon G418 incubation?

- Figure 3B is an important figure supporting one of the main biological findings of the manuscript. However, it lacks any quantitative aspect of the degree of codon enrichment and depletion among G418-sensitive transcripts. Table1, which should contain the supporting raw information for the figure lacks a header to describe what each column corresponds to. The authors should include the name of each column in the supplementary table and show a plot displaying the extent of codon enrichment and depletion among G418-sensitive transcripts compared to the mean values of codon frequency obtained from a set (of similar number) of randomly chosen transcripts among the G418-insensitive transcripts (random sampling with replacement). Authors should also indicate in the figure which codons correspond to hydrophobic amino-acids.

Minor comments:

- Authors mention in the introduction that the more prevalent codons are typically decoded by the more abundant tRNAs. Although this is the case in some bacteria species as well as in yeast and some metazoans such as C.elegans, it is not the case in mammals (this reference is a good evidence for the lack of translational selection in organisms with large genomes and a small set of tRNA coding genes https://academic.oup.com/nar/article/32/17/5036/1333956). The sentence should therefore be corrected to indicate that in some organisms, but not all, there is a correlation between codon occurrence in the transcriptome and tRNA abundance.

- The concentrations of each translation inhibitor tested in Figure 1A and Figure 1B should be mentioned in the legend or directly in the plot and not only in the Material & Methods section. The authors should also clearly confirm that the lowest translation inhibitor concentrations described in figure 1A (5ng/µl of Puromycin, 2.5ng/µl of cycloheximide and 5ng/µl of G418) are the ones used in Figure 1B.

- Figure 2B. The choice of symbols to display the p-values corresponding to the comparison of the mRNA half-lives between the different conditions tested is misleading because the number of stars is usually correlated to the p-value (p-value *>**>***). The authors should change their nomenclature and either choose a different color for each comparison made or directly display the associated p-value in the chart.

- Figure 2C. Authors should display the codon sequence corresponding to each barplot (or prepare a heat map with the sequence of each codon and the corresponding CSC-value for each condition tested). This would allow readers to clearly see if AU rich or GC rich codons are enriched among positive or negative CSCs.

Reviewer #3: Manuscript Number: PONE-D-22-00848

Full Title: Amino acid misincorporation reduces mRNA stability

Although I disagree with the author’s interpretation of their data (see major concern 1), this paper is a quality submission by an established RNA researcher and two undergraduate student co-authors. The manuscript is generally well written and provides data useful to the field. In this submission, the authors use different doses of translation inhibitors to interrogate their effect(s) on mRNA stability. The core of their argument is that the use of different doses of G418, which has been shown to increase the misincorporation of amino acids, would offer a window into how cellular RNA surveillance mechanisms would survey and ‘deal with’ RNAs where the ribosome incorporates an incorrect amino acid. The use of in vivo labeling (SLAM-Seq) is appropriate and helps make their case that the effect is at the RNA level.

Major Concerns:

1. This reviewer’s dominant concern is related to the mechanism proposed to explain the observations. I wonder why the authors invoke a protein-folding mechanism via Ubr1-CCR4/NOT complex as the method for RNA turnover. They could have a stronger case for this logic if they had data or cited papers that demonstrate that their G418 conditions yielded consistent and common misincorporation of incorrect amino acids (therefore suggesting a misfolded protein-driven mechanism). Further, in this reviewer’s opinion, the link between codon optimality and protein misfolding is tenuous and they offer no direct data or references to strengthen it.

Frankly, I think that the authors’ protein folding-targeted explanation in the discussion is not supported. In the eyes of this reviewer, the authors are wrong to neglect mentioning ribosome collisions (see many papers by the Green, Hegde, Zaher and several other labs) as the likeliest mechanism for the observed RNA instability. To this reviewer, their RNA-based codon optimality data perfectly support such a mechanism without the need to include protein folding. Further, these data fit well with previously published ribosome collision literature which show that slowing the rate of ribosome elongation (say by A-site competition or by multiple imperfect wobble pairing codons in a row) cause ribosome collisions and RNA degradation. Therefore, it could offer an important insight into another mechanism by which ribosome collisions can be studied. I urge the authors to read papers from this sub-field of RNA biology, then reconsider their data from this viewpoint, and adjust the introduction and discussion to reflect that they have considered this possibility.

As an alternate (or complimentary) course, I would also welcome a robust defense of the proposed protein folding mechanism as the cause of RNA instability.

2. In the eyes of this reviewer, the data are sound. My only question is why is some of it being held back?

For example, the authors do SLAM-seq, but they only report a small scrap of the data (limited to Fig 2b) even though half-lives were determined for over 10000 mRNAs. Why not report these data either whole or in part (limit it to mRNAs enriched in under- or over-represented codons?) as supplementary tables in this manuscript? Such data could be very useful to the broader community. At least a rationale for this omission should be offered.

3. The text, legend, and figure pertaining to figure 3 were confusing to this reviewer. It appears as if the color coding was incorrect or the descriptions are mis-assigned. Lines 490-91 state that U wobble codons are overrepresented, but the color codes show them as underrepresented. Since getting that correct is critical for interpreting the data, this needs to be corrected and the text must be adjusted to account for the changes.

4. The organization of the supplementary tables needs to be greatly improved. First, add an extra sheet as the first sheet which functions essentially as both a table of contents and as a brief summary of the worksheets in the table (since it has two sheets). Second, the tabs for each worksheet in the xls table should be titled the table. Third, the worksheets need column headings

Minor points:

1- Several citations (for lines 78,79 is just one example) are missing. Please add them as needed.

2- please break up the results section by using section headings to help organize the results by the key findings.

3- Changing figure 2C by including the actual codon identities in their codon optimality map would be VERY helpful. Currently, it’s impossible to determine which of the 61 codons is best/worst or anything in between.

4- Description of the western blot are missing in the methods. Please add

5- How was the loading of the western blots normalized? Please describe in methods and or legend.

6- The number of replicates is not listed for all experiments. Please correct.

7- For better readability, the total protein description of method (AHA/click seq) should be presented separately from the 4-thiouracil method.

8- How do we know is the number of cells remains the same and the effect is not the cells being dead or a variability of cell number.

9- For tAI experiment, please explain why the comparison was performed between puro and G418 and not with the control experiment?

10- Violin plots would be much more informative for the data presented in figures 2B, D and E.

11- Fig 2B: How many time points were used for SLAM-Seq experiments? How many replicates were performed? Although the P-value showed significant difference between Control, G418, and puromycin-treated samples, the standard deviation values were quite high.

12- Fig 1B: The authors should indicate where the primers located, are they at 3’, 5’ or CDS? (since the qPCR results may be affected by the positions of primers.)

13- Fig 1C: The authors should indicate if they used different amounts of rabbit reticulocyte lysate or antibiotics. In case different amounts of G418 were used, the authors did not explain why the firefly protein increased while the mRNA got destabilized after 30 mins reaction.

14- The authors should also provide something akin to a future direction in the discussion. For example, Selected reporter genes (ID’d by the SLAM-seq data) containing overrepresented and underrepresented codons found in fig 3 should be assayed using Northern Blotting and polysome gradient followed by Western Blotting in the condition with and without G418 to validate the conclusion that G418 acts preferentially on codons with G or C in the wobble position.

a. Please note that I’m NOT suggesting that the authors must do this for publication. Such experiments would be a great set of follow-up experiments, but in my opinion would be beyond the scope of this focused manuscript.

15- Adding 3 Vertical lines to separate the groups in Figure 3A would be very helpful (A 1,2,3 /line/ C 1,2,3 /line/ G 1,2,3 /line/ U 1,2,3). (This is best shown by C3 and G1 which look like they belong together.)

16- Bold your figure legend titles for each figure.

17- the scale on figure 3A would benefit by the inclusion of more 'ticks' and/or color-coding to show where the values are.

Vocabulary & grammar (there are several others not listed here)

- Line 62: “selected” instead of “select”

- Line 77: “A sites” instead of “A, sites”

- Line 80: not sure what binding the authors mentioned.

- Line 277: Is it ESGRO-2i medium?

- In line 463 use OR (not and) since you don't do the double inhibitor

As a final aside, this reviewer has evaluated R15 grants as a study section member. If the PI is considering applying for such a grant mechanism, then including language similar to the below in the acknowledgements section of this and every published paper from their lab would aid their application by establishing their undergraduate-focused ‘training bona fides’ for the review process.

“YTD and AS are undergraduate student trainees majoring in _____ at Washington and Lee University and were mentored by KF during this project.”

6. PLOS authors have the option to publish the peer review history of their article (what does this mean?). If published, this will include your full peer review and any attached files.

Reviewer #1: No

Reviewer #2: No

Reviewer #3: No

---

## [Author Response · Author response to Decision Letter 0]

8 Apr 2022

Editorial Comments:

As reviewers also emphasize, there is no direct evidence provided in the manuscript for link between amino acid misincorporation and mRNA stability. The observations described can be explained via alternative models. Thus, the current title and undue focus on this one mechanism is unwarranted. Either more direct evidence will be needed to support the current model, or the title and text should be extensively revised to present a more balanced interpretation of data reported in the manuscript. In any scenario, various possible modes by which translation can impact mRNA stability should be discussed.

Reviewer 3 make a key point about ribosomal collisions as another potential mechanism, and we now discuss that possibility in the revised text. We have also performed in vitro experiments to show that most luciferase mRNA in reticulocytes is bound to monoribosomes and that G418 can delay translation elongation rates (although at higher concentrations than those used to decrease mRNA stability). These data are more consistent with a model involving amino acid misincorporation, although we cannot completely rule out ribosomal collisions. We make those conclusions in the revised text and discuss both possibilities in the Discussion. Here, we do not point to specific lines of text since these changes are made throughout the manuscript.

New experiments have been suggested to obtain data for more direct support of the conclusions regarding G418 effects on mRNA stability in a codon dependent manner (reviewer 1, comments 2 and 3; reviewer 2, major comment 2). One possible way to address this could be to perform new experiments focused on select G418 sensitive and insensitive RNAs. Another possibility is to acknowledge the lack of direct evidence for such a mechanism and extensively revise the conclusions while clearly stating limitations of the data and possible alternate explanations for the observations.

Unfortunately, we are unable to perform the suggested experiments in this short time frame. As suggested above, we have both acknowledged the lack of direct evidence here. Amended text can be found in lines 256-258.

Both reviewer 1 and 2 point out the need to address the discrepancy between small effect on luciferase activity but significant impact on reporter RNA stability by intermediate levels of G418. Reviewer 2 suggests a possible source of this discrepancy that can be experimentally tested. Another way to address this issue could be to note this discrepancy with a discussion of possible caveats and limitations of the experimental approach.

Reviewer 2 suggested performing a time-course analysis for G418, which we now include in our revised manuscript. As suggested, firefly luciferase accumulates in a large burst toward the end of our time course. These data are presented in figure 1C, and we have included our interpretation of these results in lines 130-136.

Both reviewers 2 and 3 express concerns about data analysis and data presentation. All comments concerning these issues should be addressed with more thorough analysis of the available data and its appropriate presentation so that data on which conclusions are based is readily accessible to readers.

These revisions were extensive, and we respond to each comment below. In summary, we have attempted to more thoroughly analyze the data and alter its presentation (where suggested) to make it more accessible.

Reviewer #1: The article titled “Amino acid misincorporation reduces mRNA stability” by Durmaz et al., addresses an important aspect of RNA Biology. Earlier reports suggest the role of codon optimality and codon sequences in determining mRNA stability. Here the authors tried to suggest a hypothesis where the mRNA stability is affected by misincorporation of amino acid during the process of translation. In this article the authors mimicked the process of amino acid misincorporation using G418 (Geneticin). They further hypothesize that misincorporation of amino acid by ribosome leads to misfolded protein response resulting in protein degradation, where authors wanted to connect this process with mRNA stability.

The article addresses an interesting question and can be considered for publication. However the following issues need to be addressed.

Major comments:

1. G418 treatment reduced the stability of firefly luciferase mRNA in vitro (Figure 1 B). But it did not affect the protein level of firefly luciferase in Figure 1C. It is unclear why this would be the case and authors did not comment about the same.

Based on its probable mechanism, we expect G418 to have a more pronounced effect on enzyme activity than protein levels since it promotes amino acid misincorporation. Therefore, it could be the case that the modest defect in luciferase activity is not fully born out at the protein level. We have included this language in the figure legend for Fig. 1F (originally Fig. 1C) and in lines 166-169.

2. To strengthen the hypothesis proposed by author connecting G418 treatment and mRNA stability, it would be important to test and validate the stability of couple of G418 sensitive mRNA targets with specific codon features.

It is possible to perform these experiments with engineered reporter mRNAs whose transcription can be downregulated, but we are unable to do such experiments rapidly. As suggested by the editor, we have amended the text to discuss the limitations of the SLAM-Seq experiment in lines 256 – 258.

3. To confirm that amino acid mis-incorporation leads to ribosome slow-down and affect mRNA stability. An important experiment could be to express firefly luciferase mRNA containing optimal codons and non-optimal codons in vivo, check for its translation and stability.

This is an excellent experiment, although extremely challenging. We do show in our revised manuscript that G418 can delay translation elongation rates, consistent with the effect of cycloheximide. A more formal proof that codon biases lead to defects in elongation rates requires engineering new luciferase constructs to test this hypothesis. Due to time constraints, we are unable to do these additional analyses. As discussed in our response to the editor, we now more formally outline the limitations of our assays and provide these explanations as alternate interpretations of our results. The relevant lines of text are 128 – 145.

Reviewer #2: In this manuscript, Durmaz YT, Shatadal A and Friend K, studied the impact of different translation inhibitors on modulating mRNA stability in vitro and in mouse embryonic stem cells.

Using an in vitro translation system programmed with a luciferase reporter mRNA in the presence of G418 (an aminoglycoside that alters translation elongation and termination), cycloheximide (which interferes with the translocation step and blocks elongating ribosomes on the mRNA) or puromycin (an aminonucleoside that induces premature chain termination, releasing elongating ribosomes from the mRNA), the authors measured transcript stability in a 30 minutes time course. Their results indicate a significant stabilization of the reporter mRNA in the presence of cycloheximide and puromycin (compared to the untreated control), while G418 leads to accelerated mRNA degradation (compared to the untreated control).

Slam-seq experiments performed in mouse embryonic stem cells validate their in vitro results, showing a global stabilization of cellular transcripts when cells are incubated in puromycin, while incubation with G418 is associated with transcript destabilization. Calculation of codon stabilization coefficients (CSCs) using the Slam-seq datasets from control cells also recapitulates previous findings obtained in other cell-types and in particular the observed GC content bias for stabilizing and destabilizing codons described in Hia et al 2019. Interestingly, addition of puromycin or G418, although having an impact on global mRNA stability, does not appear to significantly affect the CSCs values, even though a small effect is apparent on most codons.

Finally, analysis of codon usage on G418-sensitive mRNAs revealed an enrichment of codons with G or C in the wobble position, which are typically stabilizing codons. Furthermore, codons under-represented in the G418-sensitive mRNAs correspond to hydrophobic amino-acids.

Based on their results and in ribosome crystal structures obtained in the presence of G418, the authors conclude that G418 could preferentially affect near-cognate tRNA usage in GC rich codons leading to increase amino-acid mis-incorporation and inducing mRNA degradation.

Overall, the manuscript is very well written. The introduction is well documented and relevant to the study as it allows readers to place the study in the current context of the field while highlighting open questions that have not been addressed yet. The authors have generate an interesting dataset using the state-of-the-art Slam-seq protocol. However, the analyses performed on this dataset are sometimes cryptic and superficial and could be largely improved. Moreover, there is room for improving the overall clarity of the figures to facilitate the interpretation of the results by readers.

Below you will find some major and minor points that, in my opinion, should be addressed by the authors.

Major points:

- I am surprised that the low concentrations of translation inhibitors used in the in vitro time course (Figure 1B), have such a profound impact on the reporter mRNA stability while leading to a decrease of less than 10% of the luciferase activity at 30 minutes as shown in Figure 1A. This could be due to the accumulation of most of the luciferase protein during the first minutes of the translation reaction followed by a plateau in its abundance at later points. It would therefore be important for the authors to include a new time-course figure where firefly luciferase is measured for each sample at each time-point so readers can compare the dynamics of mRNA levels and protein accumulation.

As discussed in response to Reviewer 3, we have now included the suggested time course.

- Authors decided to perform the Slam-seq protocol in the presence of small amounts of G418 or Puromycin which have modest effects on total translation rates (20% inhibition based on the results presented in Figure 2A). It is therefore not surprising that the effects obtained are very mild. Although translation is an important determinant of mRNA stability, it is probable that only a small fraction of all translating ribosomes are involved in inducing mRNA degradation. It would be interesting to include an additional experiment using a higher concentration of G418 in the analysis to obtain a clearer and robust picture of the impact of G418 on codon-dependent mRNA stability. I also think it would be important to perform a sucrose gradient experiment at the different doses of translation inhibitors to look at their effect on the polysome profile.

In any case, contrary to what it is stated in the results section, I do not think (at least for puromycin) that the effect of the translational inhibitors on modulating mRNA stability are independent from codon optimality. There is robust evidence from the work of Olivia Rissland and Ariel Bazzini laboratories showing that translation inhibition leads to a loss of the effect of codon-optimality on mRNA stability.

This is a great point, and you are right that we used lower translation inhibitor concentrations in our study. Our goal was to inhibit translation, but not to such a degree that cells would senesce over the course of the experiment. Reanalyzing the data with higher concentrations of inhibitors would likely lead to off-target and nonspecific effects.

To better understand ribosome dynamics in the presence of the inhibitors, we have performed polysome analysis, but using our in vitro system. We have also further characterized that system where it is possible to add a variety of translation inhibitor concentrations with less concern about off-target effects. Those results are reported in Figs. 1C – E and S1. The relevant figures are discussed in lines 128-160. 

- Figure 2B: Instead of displaying a bar plot with the half-lives of cellular transcripts, it would be preferable to use a density plot so readers can have a clearer view of the distribution of half-lives in each condition.

Reviewer #3 suggested violin plots, so we have altered Figure 2, panels B, D, and E appropriately.

- G418-sensitive transcripts were obtained by calculating the ratio of mRNA half-lives in the G418 treated samples against Puromycin-treated samples (as described in the Material and Methods section). I supposed this was done to maximize the difference in half-lives since the effect of each drug are mild compared to the control condition. However, this strategy might introduce a bias as puromycin also has a small effect on CSC values which are different from that of G418. In my opinion it would be preferable to use the control condition to identify G418-sensitive transcripts. I also think that authors should show a plot with the distribution of the obtained ratios and the two-fold cutoff chosen to define G418-sensitive transcripts.

Our goal with using the puromycin treatment was to look at a condition where translation inhibition was stabilizing, rather than destabilizing. Many factors regulate mRNA stability, so with this comparison, we desired to focus on translation itself. That being said, the point is well-taken. We have revised the criteria defining G418-sensitive mRNAs to now include control reactions. In summary, we defined G418-sensitive mRNAs as those whose mRNA stabilities are highest in puromycin-treated cells, intermediate in control cells and lower in G418-treated cells. We still use the two-fold difference in mRNA half-lives between puromycin-treated and G418-treated cells as a final cutoff. By increasing the stringency of our cutoffs, we reduced the overall number of G418-sensitive mRNAs, but did observe trends in codon bias (see Fig. 3B) consistent with our earlier results. Our new criteria are discussed in lines 217 – 220 and 419 – 21.

- Are G418-sensitive transcripts enriched in a specific functional category? Are transcripts with the longer open reading frames more sensitive to G418 since they will potentially accumulate more mis-incorporated amino-acids in their nascent chains? Is the Ribosome-associated quality control machinery recruited to polysomes upon G418 incubation?

These are excellent questions. No, G418-sensitive transcripts are not enriched for certain biological processes or categories. They also do not contain longer or shorter ORFs on average. Those data are presented in lines 229 - 232. Examining the ribosome-associated quality control machinery would require extensive additional experiments. In consideration of the comments by reviewer #3, we present a more detailed discussion about this possibility.

- Figure 3B is an important figure supporting one of the main biological findings of the manuscript. However, it lacks any quantitative aspect of the degree of codon enrichment and depletion among G418-sensitive transcripts. Table1, which should contain the supporting raw information for the figure lacks a header to describe what each column corresponds to. The authors should include the name of each column in the supplementary table and show a plot displaying the extent of codon enrichment and depletion among G418-sensitive transcripts compared to the mean values of codon frequency obtained from a set (of similar number) of randomly chosen transcripts among the G418-insensitive transcripts (random sampling with replacement). Authors should also indicate in the figure which codons correspond to hydrophobic amino-acids.

We have included these analyses as supplemental table 3, providing the averaged values for each codon and the corresponding statistics comparing G418-sensitive versus insensitive mRNAs, and we have revised tables to contain headings as suggested. As suggested, Fig. 3B has also been revised.

Minor comments:

- Authors mention in the introduction that the more prevalent codons are typically decoded by the more abundant tRNAs. Although this is the case in some bacteria species as well as in yeast and some metazoans such as C.elegans, it is not the case in mammals (this reference is a good evidence for the lack of translational selection in organisms with large genomes and a small set of tRNA coding genes https://academic.oup.com/nar/article/32/17/5036/1333956). The sentence should therefore be corrected to indicate that in some organisms, but not all, there is a correlation between codon occurrence in the transcriptome and tRNA abundance.

We have made this correction, including the indicated refernce. See line 74.

- The concentrations of each translation inhibitor tested in Figure 1A and Figure 1B should be mentioned in the legend or directly in the plot and not only in the Material & Methods section. The authors should also clearly confirm that the lowest translation inhibitor concentrations described in figure 1A (5ng/µl of Puromycin, 2.5ng/µl of cycloheximide and 5ng/µl of G418) are the ones used in Figure 1B.

We have amended the figure legend to explicitly include this information.

- Figure 2B. The choice of symbols to display the p-values corresponding to the comparison of the mRNA half-lives between the different conditions tested is misleading because the number of stars is usually correlated to the p-value (p-value *>**>***). The authors should change their nomenclature and either choose a different color for each comparison made or directly display the associated p-value in the chart.

We have made this change as well.

- Figure 2C. Authors should display the codon sequence corresponding to each barplot (or prepare a heat map with the sequence of each codon and the corresponding CSC-value for each condition tested). This would allow readers to clearly see if AU rich or GC rich codons are enriched among positive or negative CSCs.

We have included this information in Figure 2C and provide color coding for wobble positions to aid the reader.

Reviewer #3: Manuscript Number: PONE-D-22-00848

Full Title: Amino acid misincorporation reduces mRNA stability

Although I disagree with the author’s interpretation of their data (see major concern 1), this paper is a quality submission by an established RNA researcher and two undergraduate student co-authors. The manuscript is generally well written and provides data useful to the field. In this submission, the authors use different doses of translation inhibitors to interrogate their effect(s) on mRNA stability. The core of their argument is that the use of different doses of G418, which has been shown to increase the misincorporation of amino acids, would offer a window into how cellular RNA surveillance mechanisms would survey and ‘deal with’ RNAs where the ribosome incorporates an incorrect amino acid. The use of in vivo labeling (SLAM-Seq) is appropriate and helps make their case that the effect is at the RNA level.

Major Concerns:

1. This reviewer’s dominant concern is related to the mechanism proposed to explain the observations. I wonder why the authors invoke a protein-folding mechanism via Ubr1-CCR4/NOT complex as the method for RNA turnover. They could have a stronger case for this logic if they had data or cited papers that demonstrate that their G418 conditions yielded consistent and common misincorporation of incorrect amino acids (therefore suggesting a misfolded protein-driven mechanism). Further, in this reviewer’s opinion, the link between codon optimality and protein misfolding is tenuous and they offer no direct data or references to strengthen it.

Frankly, I think that the authors’ protein folding-targeted explanation in the discussion is not supported. In the eyes of this reviewer, the authors are wrong to neglect mentioning ribosome collisions (see many papers by the Green, Hegde, Zaher and several other labs) as the likeliest mechanism for the observed RNA instability. To this reviewer, their RNA-based codon optimality data perfectly support such a mechanism without the need to include protein folding. Further, these data fit well with previously published ribosome collision literature which show that slowing the rate of ribosome elongation (say by A-site competition or by multiple imperfect wobble pairing codons in a row) cause ribosome collisions and RNA degradation. Therefore, it could offer an important insight into another mechanism by which ribosome collisions can be studied. I urge the authors to read papers from this sub-field of RNA biology, then reconsider their data from this viewpoint, and adjust the introduction and discussion to reflect that they have considered this possibility.

As an alternate (or complimentary) course, I would also welcome a robust defense of the proposed protein folding mechanism as the cause of RNA instability.

This was an incredibly valuable observation and prompted us to ask some key questions using our in vitro translation system. While imperfect, most translation seems to occur in reticulocyte lysate on mRNA bound to monoribosomes (Figs. 1D and E). Interestingly, G418 can delay translation elongation rates at higher concentrations (Fig. S1), but does not push luciferase mRNA into heavier complexes with additional ribosomes. Furthermore, minimal luciferase mRNA can be found in denser fractions from a polyribosome sedimentation assay. These data are largely inconsistent with a ribosome collision model, but we cannot rule that possibility out.

In our revised manuscript, we cannot and do not insist on amino acid misincorporation as the sole mechanism by which G418 can act. We include the references on ribosome collisions and attempt to take a more nuanced view. These edits required extensive changes to the text and additional figures, but the key changes can be found in lines 146 – 160 and 310 – 315.

2. In the eyes of this reviewer, the data are sound. My only question is why is some of it being held back?

For example, the authors do SLAM-seq, but they only report a small scrap of the data (limited to Fig 2b) even though half-lives were determined for over 10000 mRNAs. Why not report these data either whole or in part (limit it to mRNAs enriched in under- or over-represented codons?) as supplementary tables in this manuscript? Such data could be very useful to the broader community. At least a rationale for this omission should be offered.

These data were included in our original GEO submission and can be found in that repository, but for ease-of-access, we have also included calculated mRNA half-lives as an additional supplemental table here.

3. The text, legend, and figure pertaining to figure 3 were confusing to this reviewer. It appears as if the color coding was incorrect or the descriptions are mis-assigned. Lines 490-91 state that U wobble codons are overrepresented, but the color codes show them as underrepresented. Since getting that correct is critical for interpreting the data, this needs to be corrected and the text must be adjusted to account for the changes.

We have attempted to modify this figure in response to both this comment and comments from Reviewer #2. There was some discrepancy between the text and figure.

4. The organization of the supplementary tables needs to be greatly improved. First, add an extra sheet as the first sheet which functions essentially as both a table of contents and as a brief summary of the worksheets in the table (since it has two sheets). Second, the tabs for each worksheet in the xls table should be titled the table. Third, the worksheets need column headings

We have made these changes in the supplemental tables and apologize for their original lack of clarity.

Minor points:

In the interests of brevity, we point to individual figures or lines of text where the suggested corrections are made, but we have attempted to address each point raised below.

1- Several citations (for lines 78,79 is just one example) are missing. Please add them as needed.

2- please break up the results section by using section headings to help organize the results by the key findings. The Results have been divided as suggested.

3- Changing figure 2C by including the actual codon identities in their codon optimality map would be VERY helpful. Currently, it’s impossible to determine which of the 61 codons is best/worst or anything in between. Codon sequences are included and color-coded to aid the reader.

4- Description of the western blot are missing in the methods. Please add. See lines 371 - 375.

5- How was the loading of the western blots normalized? Please describe in methods and or legend.

6- The number of replicates is not listed for all experiments. Please correct.

7- For better readability, the total protein description of method (AHA/click seq) should be presented separately from the 4-thiouracil method. This paragraph has been given its own heading.

8- How do we know is the number of cells remains the same and the effect is not the cells being dead or a variability of cell number.

9- For tAI experiment, please explain why the comparison was performed between puro and G418 and not with the control experiment? This point was raised by reviewer 2 and is addressed above.

10- Violin plots would be much more informative for the data presented in figures 2B, D and E. These are now included.

11- Fig 2B: How many time points were used for SLAM-Seq experiments? How many replicates were performed? Although the P-value showed significant difference between Control, G418, and puromycin-treated samples, the standard deviation values were quite high. These additional pieces of information are now included in the Methods. See line 397.

12- Fig 1B: The authors should indicate where the primers located, are they at 3’, 5’ or CDS? (since the qPCR results may be affected by the positions of primers.) They are located in the CDS in the case of luciferase, CFP (new to this revision) and toward the 3’end of the 18S rRNA (see lines 364 – 367).

13- Fig 1C: The authors should indicate if they used different amounts of rabbit reticulocyte lysate or antibiotics. In case different amounts of G418 were used, the authors did not explain why the firefly protein increased while the mRNA got destabilized after 30 mins reaction. This point was raised by reviewer #1, and we address it above.

14- The authors should also provide something akin to a future direction in the discussion. For example, Selected reporter genes (ID’d by the SLAM-seq data) containing overrepresented and underrepresented codons found in fig 3 should be assayed using Northern Blotting and polysome gradient followed by Western Blotting in the condition with and without G418 to validate the conclusion that G418 acts preferentially on codons with G or C in the wobble position.

a. Please note that I’m NOT suggesting that the authors must do this for publication. Such experiments would be a great set of follow-up experiments, but in my opinion would be beyond the scope of this focused manuscript. We now include a more detailed future directions section within the Discussion in lines 310 - 316.

15- Adding 3 Vertical lines to separate the groups in Figure 3A would be very helpful (A 1,2,3 /line/ C 1,2,3 /line/ G 1,2,3 /line/ U 1,2,3). (This is best shown by C3 and G1 which look like they belong together.) We have amended the figure as suggested.

16- Bold your figure legend titles for each figure. This edit has been made.

17- the scale on figure 3A would benefit by the inclusion of more 'ticks' and/or color-coding to show where the values are. We have amended the figure as suggested.

Vocabulary & grammar (there are several others not listed here)

- Line 62: “selected” instead of “select” 

- Line 77: “A sites” instead of “A, sites” 

- Line 80: not sure what binding the authors mentioned. 

- Line 277: Is it ESGRO-2i medium?

- In line 463 use OR (not and) since you don't do the double inhibitor 

We have amended these lines for clarity.

As a final aside, this reviewer has evaluated R15 grants as a study section member. If the PI is considering applying for such a grant mechanism, then including language similar to the below in the acknowledgements section of this and every published paper from their lab would aid their application by establishing their undergraduate-focused ‘training bona fides’ for the review process.

“YTD and AS are undergraduate student trainees majoring in _____ at Washington and Lee University and were mentored by KF during this project.”

We have included such language in the Acknowledgements, and we thank the reviewer for this suggestion.

---

## [Decision Letter · Decision Letter 1]

23 Jun 2022

PONE-D-22-00848R1Geneticin reduces mRNA stabilityPLOS ONE

Dear Dr. Friend,

Thank you for submitting your manuscript to PLOS ONE. After careful consideration, we feel that it needs few minor changes before it can fully meet PLOS ONE’s publication criteria. Therefore, we invite you to submit a revised version of the manuscript that addresses the points raised during the review process.

All changes requested by reviewers 1 and 2 can be made by changing the text to acknowledge caveats or add explanations as suggested. No new experiments will be necessary. Once the revised manuscript is submitted,  an additional round of peer review may not be needed and the decision can be made at the editorial level.

We look forward to receiving your revised manuscript.

Kind regards,

Guramrit Singh

Academic Editor

PLOS ONE

Journal Requirements:

Reviewers' comments:

Reviewer's Responses to Questions

**Comments to the Author**

1. If the authors have adequately addressed your comments raised in a previous round of review and you feel that this manuscript is now acceptable for publication, you may indicate that here to bypass the “Comments to the Author” section, enter your conflict of interest statement in the “Confidential to Editor” section, and submit your "Accept" recommendation.

Reviewer #1: All comments have been addressed

Reviewer #2: All comments have been addressed

Reviewer #3: All comments have been addressed

2. Is the manuscript technically sound, and do the data support the conclusions?

Reviewer #1: Yes

Reviewer #2: Partly

Reviewer #3: Yes

3. Has the statistical analysis been performed appropriately and rigorously? 

Reviewer #1: Yes

Reviewer #2: Yes

Reviewer #3: Yes

4. Have the authors made all data underlying the findings in their manuscript fully available?

Reviewer #1: Yes

Reviewer #2: Yes

Reviewer #3: Yes

5. Is the manuscript presented in an intelligible fashion and written in standard English?

Reviewer #1: Yes

Reviewer #2: Yes

Reviewer #3: Yes

6. Review Comments to the Author

Reviewer #1: Reviewer #1:

Durmaz et al has satisfactorily address all the queries raised by this reviewer either by providing the justification to our comments or by mentioning the caveats and limitations of the experimental approach within the text. Therefore, considering the scope of the focused manuscript this reviewer recommends publication in the journal.

However, this reviewer has some further questions that requires justifications.

Major comments:

1. Line 158, figure number 1C-D, the polyribosome sedimentation was carried out after inhibiting the translation reaction at 15 minutes to trap the mRNAs in the initial phase of firefly luciferase production. This reviewer has no objection about the time points taken. Reviewer is just curious that the difference of mRNA degradation in G418 vs untreated (Figure 1B) is very prominent at t=20 and 30, then why the authors decided to go with 15 minutes of translation reaction followed by polyribosome sedimentation. The authors mention in the line 108-110, that “in vitro, we observe that G418 likely acts independently of ribosome collisions.” What if the timepoints of t=20 or 30 minutes could have roughly provided us the connection between G418 and ribosome collision. Without which the above statement remains void and should be re-written.

2. The sedimentation assay (Figure 1D and 1E) was carried out to check the relative abundance of luciferase mRNA throughout the polysome fraction. From the result it was evident that the mRNA is present mostly in the monoribosomal fraction. But the authors did not mention the proper method to quantify the mRNAs. Did the author took ribosomal rRNA to measure relative abundance of luciferase mRNA? Or is it relative to each fraction of the polysome profile? It would be very informative if authors can describe this information in the method section.

Minor comments:

1. Incorporate the G418 (Geneticin) in the introduction as the title suggest and it would be informative to a novice reader.

2. Line 233-234: repetitive statement.

3. Interspersed grammatical errors. A thorough reading of the manuscript should take care of it.

Reviewer #2: Dear authors,

Thank you for having addressed most of my comments. I only have two final comments that I think should be addressed:

- Introduction: It is important to clearly state in which organism the findings described in the introduction were obtained as there can be mechanistic differences between yeast, insects and mammals. As an example, the work describing the role of the Ccr4-Not complex at ribosomes with unoccupied A sites was performed in yeast.

- Line 149-153 (Figure S1) : The authors tested whether G418 could delay elongation in RRL by monitoring luciferase expression levels (upon adding high concentrations of G418 and other translation inhibitors) using western-blotting. However, G418 induces amino-acid misincorporation and thus (at high concentrations), could possibly change the epitope recognized by the luciferase antibody. A better proxy to monitor this would be to perform in vitro translation experiments in the presence of S35-Methionine in order to quantify luciferase expression more accurately.

Reviewer #3: I commend the authors for submitting a greatly improved manuscript. They have de-emphasized and/or removed the unsupported text and have satisfactorily addressed this reviewer's core concerns.

While the authors have not definitively identified the mechanism by which G418 reduces the stability of the reporter RNAs (which is beyond the scope of this manuscript -as far as this reviewer is concerned-) they have effectively ruled out certain possible mechanisms.

On that note, I commend the authors for their use of polysome gradients to address my suggestion of ribosome collisions as the likely mechanism. Their data show nearly a complete absence of polysomes and a strong monosome peak. Simply, ribosomes can't collide when each mRNA is only translated by a single ribosome. It's a clever way to test my suggestion. As they show their polysome data do capture the regions of the gradient that span from 40S peak through part of the polysome region are convincing enough for this reviewer, they aren't perfect. (see below)

My lab regularly runs polysome gradients (using the same sucrose concentrations, and rotor) as part of our work. Simply, by this reviewer's experience their centrifugation conditions (10-50% gradients, centrifuged @ 39,000 RPM for 3 hours in an SW-41 rotor) likely resulted in the heaviest ribosomes pelleting. My lab uses very similar conditions, but we only spin samples for 2 hours (not 3). Granted, my lab's polysome results are all based on cell lysates as opposed to an in vitro translation system & there are differences between systems, but I'm fairly certain that the author's data only includes the polysome region corresponding to the position where polysomes corresponding to maybe the 5 or 6 ribosomes would be. The very long distance between the RNP peak at the start of the gradient and the monosome peak speaks to this as well.

Despite this concern, as I said above, I find these data (especially the absence of any di-ribosome peak -which, had it been there, would have been captured in their experiment) convincing enough to support publication of this manuscript.

On a final note, I also encourage the authors to submit higher quality figures for publication. They are quite pixellated on my display and when I printed them out on hardcopy.

7. PLOS authors have the option to publish the peer review history of their article (what does this mean?). If published, this will include your full peer review and any attached files.

Reviewer #1: No

Reviewer #2: No

Reviewer #3: No

---

## [Author Response · Author response to Decision Letter 1]

30 Jun 2022

We greatly appreciate the feedback from Reviewers 1-3 on this second consideration of our manuscript. We have attempted to address each reviewer’s comments in the indicated sections of our manuscript (see below, italicized). Our manuscript has been greatly improved by your careful reading and critical feedback. We greatly appreciate the thoughtfulness of your comments.

From the Editor:

All changes requested by reviewers 1 and 2 can be made by changing the text to acknowledge caveats or add explanations as suggested. No new experiments will be necessary. Once the revised manuscript is submitted, an additional round of peer review may not be needed and the decision can be made at the editorial level.

As indicated above, we have modified the manuscript in an attempt to address the remaining concerns of each reviewer. We have indicated the changes made below.

From Reviewer 1:

Durmaz et al has satisfactorily address all the queries raised by this reviewer either by providing the justification to our comments or by mentioning the caveats and limitations of the experimental approach within the text. Therefore, considering the scope of the focused manuscript this reviewer recommends publication in the journal.

However, this reviewer has some further questions that requires justifications.

Major comments:

1. Line 158, figure number 1C-D, the polyribosome sedimentation was carried out after inhibiting the translation reaction at 15 minutes to trap the mRNAs in the initial phase of firefly luciferase production. This reviewer has no objection about the time points taken. Reviewer is just curious that the difference of mRNA degradation in G418 vs untreated (Figure 1B) is very prominent at t=20 and 30, then why the authors decided to go with 15 minutes of translation reaction followed by polyribosome sedimentation. The authors mention in the line 108-110, that “in vitro, we observe that G418 likely acts independently of ribosome collisions.” What if the timepoints of t=20 or 30 minutes could have roughly provided us the connection between G418 and ribosome collision. Without which the above statement remains void and should be re-written.

In performing the experiment this way, our chief objective was to interrogate ribosome-bound mRNAs prior to significant differences in mRNA levels. We selected the 15 min time-point since that time allowed ribosome loading onto mRNAs, but was likely concurrent with their initial decay. As the reviewer notes, between 10 and 20 min (and before to some degree), we observe reduction in mRNA levels implying that mRNAs are decaying within that time window. We have updated the text to make this point more explicitly (see lines 158 - 59).

2. The sedimentation assay (Figure 1D and 1E) was carried out to check the relative abundance of luciferase mRNA throughout the polysome fraction. From the result it was evident that the mRNA is present mostly in the monoribosomal fraction. But the authors did not mention the proper method to quantify the mRNAs. Did the author took ribosomal rRNA to measure relative abundance of luciferase mRNA? Or is it relative to each fraction of the polysome profile? It would be very informative if authors can describe this information in the method section.

As indicated in the revised Methods, normalization was accomplished using a spike-in mRNA since various polysome fractions have different levels of rRNA. The Methods are changed in line 363 to indicate this point.

Minor comments:

We have made the changes in the text where indicated below.

1. Incorporate the G418 (Geneticin) in the introduction as the title suggest and it would be informative to a novice reader. (lines 103 - 107)

2. Line 233-234: repetitive statement. Removed.

3. Interspersed grammatical errors. A thorough reading of the manuscript should take care of it.

Edited throughout.

From Reviewer 2:

- Introduction: It is important to clearly state in which organism the findings described in the introduction were obtained as there can be mechanistic differences between yeast, insects and mammals. As an example, the work describing the role of the Ccr4-Not complex at ribosomes with unoccupied A sites was performed in yeast.

This is an excellent point. We have attempted to specify which systems were used to make the observations in the Introduction. Changes are indicated throughout using track changes.

- Line 149-153 (Figure S1) : The authors tested whether G418 could delay elongation in RRL by monitoring luciferase expression levels (upon adding high concentrations of G418 and other translation inhibitors) using western-blotting. However, G418 induces amino-acid misincorporation and thus (at high concentrations), could possibly change the epitope recognized by the luciferase antibody. A better proxy to monitor this would be to perform in vitro translation experiments in the presence of S35-Methionine in order to quantify luciferase expression more accurately.

Absolutely, labeling with 35S-Met would be ideal. However, we do not currently have a phosphorimager or access to a dark room for X-ray film development. That necessitates the use of antibodies or firefly luciferase activity measurements (which were used and are now indicated the in Fig. S1 legend). That being said, the activity that we do observe obeys delayed kinetics. The point is certainly valid, however, so we have modified the text in lines 149 - 51 to discuss this caveat and to be more explicit about how firefly luciferase protein levels were monitored. 

From Reviewer 3:

I commend the authors for submitting a greatly improved manuscript. They have de-emphasized and/or removed the unsupported text and have satisfactorily addressed this reviewer's core concerns.

While the authors have not definitively identified the mechanism by which G418 reduces the stability of the reporter RNAs (which is beyond the scope of this manuscript -as far as this reviewer is concerned-) they have effectively ruled out certain possible mechanisms.

On that note, I commend the authors for their use of polysome gradients to address my suggestion of ribosome collisions as the likely mechanism. Their data show nearly a complete absence of polysomes and a strong monosome peak. Simply, ribosomes can't collide when each mRNA is only translated by a single ribosome. It's a clever way to test my suggestion. As they show their polysome data do capture the regions of the gradient that span from 40S peak through part of the polysome region are convincing enough for this reviewer, they aren't perfect. (see below)

My lab regularly runs polysome gradients (using the same sucrose concentrations, and rotor) as part of our work. Simply, by this reviewer's experience their centrifugation conditions (10-50% gradients, centrifuged @ 39,000 RPM for 3 hours in an SW-41 rotor) likely resulted in the heaviest ribosomes pelleting. My lab uses very similar conditions, but we only spin samples for 2 hours (not 3). Granted, my lab's polysome results are all based on cell lysates as opposed to an in vitro translation system & there are differences between systems, but I'm fairly certain that the author's data only includes the polysome region corresponding to the position where polysomes corresponding to maybe the 5 or 6 ribosomes would be. The very long distance between the RNP peak at the start of the gradient and the monosome peak speaks to this as well.

Despite this concern, as I said above, I find these data (especially the absence of any di-ribosome peak -which, had it been there, would have been captured in their experiment) convincing enough to support publication of this manuscript.

Although not specifically requested, we do now point out that our polysome profiles may exclude very large complexes of ribosomes and mRNA given how we prepared the samples. Those changes are made in lines 164 - 6.

On a final note, I also encourage the authors to submit higher quality figures for publication. They are quite pixellated on my display and when I printed them out on hardcopy.

We have attempted to upload higher resolution images.

---

## [Editor Report · Decision Letter 2]

13 Jul 2022

Geneticin reduces mRNA stability

PONE-D-22-00848R2

Dear Dr. Friend,

We’re pleased to inform you that your manuscript has been judged scientifically suitable for publication and will be formally accepted for publication once it meets all outstanding technical requirements.

Kind regards,

Guramrit Singh

Academic Editor

PLOS ONE

Additional Editor Comments (optional):

All remaining reviewer concerns are now adequately addressed and manuscript is now acceptable for publication. 
---

## [Editor Report · Acceptance letter]

18 Jul 2022

PONE-D-22-00848R2 

Geneticin reduces mRNA stability 

Dear Dr. Friend:

I'm pleased to inform you that your manuscript has been deemed suitable for publication in PLOS ONE. Congratulations! Your manuscript is now with our production department. 

Kind regards, 

on behalf of

Dr. Guramrit Singh 

Academic Editor

PLOS ONE